# Understanding the perspectives of people with dementia and family carers about clinical pharmacists in primary care: A qualitative study

Alice Burnand[1]*, Abi Woodward[2], Kumud Kantilal[1], Cini Bhanu[1], Yogini Jani[3,4], Jill Manthorpe[5,6], Mine Orlu[4,7], Greta Rait[1,8], Madiha Sajid[1], Kritika Samsi[5,6], Victoria Vickerstaff[9], Jane Ward[1], Jane Wilcock[1], Nathan Davies[2]

1 Research Department of Primary Care and Population Health, Centre for Ageing Population Studies, Patient and Public Involvement Member, University College London, London, United Kingdom, 2 Centre for Psychiatry and Mental Health, Wolfson Institute of Population Health, QMUL, London, United Kingdom, 3 Research Department of Practice and Policy, School of Pharmacy, University College London, London, United Kingdom, 4 Centre for Medicines Optimisation Research and Education, University College London Hospitals NHS Foundation, London United Kingdom, 5 NIHR Policy Research Unit in Health and Social Care Workforce, King's College London, London, United Kingdom, 6 NIHR Applied Research Collaborative (ARC) South London, King's College London, London, United Kingdom, 7 Research Department of Pharmaceutics, UCL School of Pharmacy, University College London, London, United Kingdom, 8 Research Department of Primary Care and Population Health, PRIMENT Clinical Trials Unit, University College London, London, United Kingdom, 9 Centre for Evaluation and Methods, Wolfson Institute of Population Health, QMUL, London, United Kingdom

* a.burnand@ucl.ac.uk

## Abstract

### Background

The number of people living with dementia is increasing, placing significant strain on healthcare systems and family carers. Primary care teams, including clinical pharmacists, are crucial in supporting people with dementia. While clinical pharmacists have demonstrated benefits in other areas of healthcare, their role in dementia care is less understood. This study aims to explore the perspectives of people with dementia and family carers on the potential contributions of clinical pharmacists to dementia support within primary care in England.

### Objectives

To explore the views and perspectives of people with dementia and family carers of dementia care received from primary care teams, with a specific focus on clinical pharmacists in England.

### Methods

We conducted thirteen semi-structured interviews with family carers and fifteen with people with dementia in 2022–2024. Interviews were analysed using reflexive thematic analysis.

**Data availability statement:** All relevant data are within the manuscript and its Supporting information files.

**Funding:** This study/project is funded by the National Institute for Health and Care Research (NIHR) School for Primary Care Research (project reference 575). The views expressed are those of the authors and not necessarily those of the NIHR or the Department of Health and Social Care. The funders had and will not have a role in study design, data collection and analysis, decision to publish, or preparation of the manuscript. There was no additional external funding received for this study.

**Competing interests:** The authors have no competing interests to declare.

## Results

Three overarching themes, were developed from the interviews: 1) Recognising the value of clinical pharmacists 2) Building and developing rapport, with personalised care and a holistic approach; and 3) The needs of patients without clinical pharmacy support – left in uncertainty about their care.

## Conclusion

This study highlights the potential of clinical pharmacists in dementia care, highlighting both positive experiences as well as unmet needs of those who did not have access to the service. Awareness of these services hinder the impact that they might be able to achieve otherwise. Increasing awareness, standardising training, and further research on service delivery models are crucial.

## Introduction

There are over 57 million people living with dementia worldwide [1] and this is expected to increase to 131.5 million by 2050 [2]. The majority of care for a person with dementia is provided by family carers, which may also include friends [3]. For most patients in health care systems in high income countries, primary care is the first point of contact for health-related concerns, including dementia [4]. The organisation and composition of primary care teams varies internationally. In the UK, primary care teams may include professionals such as GPs, nurses, physiotherapists, paramedics, health and social care professionals and clinical pharmacists [5].

As the number of people with dementia increases [2], both specialist care models and primary care teams will need to adapt to meet growing needs. Significant workload pressures already affect GPs, which has a direct impact the quality of patient care [6], and with the number of full-time equivalent GPs declining [7,8], multidisciplinary support within primary care is urgently needed.

Clinical pharmacists are a relatively new role within the National Health Service (NHS) in England. The NHS in the UK is a service funded through national taxation, with its framework set by the government [9]. Since the 1980s, pharmacists have increasingly provided direct patient care [10], and clinical responsibilities for those working in general practice has expanded in recent years [11,12]. Clinical pharmacists have a distinct role from that of community pharmacists, who are primarily involved in dispensing medications in retail pharmacy settings [13]. Clinical pharmacist duties include medication optimisation, such as prescribing and deprescribing medication [14], providing they have undertaken the relevant independent prescribing training available to pharmacists in the UK [15].

Clinical pharmacists' contribution to primary care and wider healthcare systems is well documented [10,16,17]. They have been shown to reduce NHS appointment waiting times, enhance patient safety, reduce medication wastage and overuse, and increase access to healthcare, particularly for those with multiple long-term conditions

[18–21]. Clinical pharmacists work within the primary care teams, alongside GPs and consult with patients directly through the GP surgery or Primacy Care Networks (PCNs) [22]. However, with the recent shift towards patient-facing duties for clinical pharmacists, there is limited research into understanding patients' and carers' perspectives of their evolving role [23].

A recent study utilising focus groups of patients in the general population found low awareness and confusion about the pharmacist role but reported high acceptability and a positive impact on medication use [24]. Moreover, carers of people with dementia have reported they receive limited information on medication from health care professionals during key transitions, such as hospital discharge [25]. Clinical pharmacist with their specialist knowledge and expertise are well placed to support complex cases, including older adults with multiple long-term conditions such as dementia. However, little is known about the role clinical pharmacists can play in supporting people with dementia and family carers in the community [23].

The support and care needs of people with dementia, typically provided by family carers, increases with advancing disease [26]. Family carers are instrumental in contributing to positive outcomes and quality of life for people with dementia, as evidenced by lower hospitalisation rates [27] and delayed institutionalisation [28]. While family carers are invaluable, they cannot meet all care needs, and in some instances, people with dementia may not have a family network, paid, or unpaid support.

Effective health and social care services are essential to address care needs, including therapeutic interventions and medications management. Increasing age and presence of long-term conditions such as dementia, are both linked to polypharmacy which requires specialist support and input [29]. Extended periods of polypharmacy are associated with increased hospitalisation and mortality in people with dementia [30]. Clinical pharmacists are well positioned to enhance medication safety and support care for people with dementia [28,31], however little is known about how they can best support people with dementia and their family carers. Our study aimed to explore from the perspective of people with dementia and family carers how clinical pharmacy services can support them.

### Aims and objectives

This study aimed to explore with people with dementia and family carers their views and experience of dementia care from primary care teams, with a specific focus on clinical pharmacists in England.

## Materials and methods

### Research design

Semi-structured interviews with people with dementia and family carers, analysed using reflexive thematic analysis [32], and reported using the COREQ checklist [33].

### Recruitment and sample

Participants were recruited using purposive sampling with an aim to recruit a diverse sample including age, sex, and ethnicity. We monitored for these criteria throughout recruitment to ensure a range of characteristics as far as possible. For carers, this included being the primary carer for someone with a dementia diagnosis, residing in England. For people with dementia, this involved a confirmed diagnosis of any type of dementia and residing in England. Both groups were recruited across different ethnicities, sex and age groups. Our sample size was guided by the principles of information power, which considers the study aim, sample specificity, quality of dialogue and analysis strategy [34]. We anticipated recruiting approximately 15 people with dementia and 15 family carers. Recruitment started on 01/07/2022 and finished on 01/04/2024.

Participants were recruited through general practice, charities, NHS networks, and a service that connects registered volunteers with dementia researchers in the UK (Join Dementia Research) via telephone and email. The study was also

advertised on social media (LinkedIn and X (formerly known as Twitter)), using posters and flyers. Snowball sampling methods and known contacts of the research team supported the recruitment process. Four GP practices located in London supplemented recruitment. Practices identified people with dementia and carers registered on their databases. Prospective participants were sent information sheets and consent forms and invited to contact the research team using a reply slip with their contact details. The research team then contacted the prospective participants with further study information. All participants were required to provide informed consent (written or oral) using an informed consent form.

The inclusion criteria for people with dementia were a clinical diagnosis of any type of dementia and receiving any medication-related support from a primary care service, including a clinical pharmacist. Lack of capacity and inability to give informed consent excluded participation.

At the start of the project, we required people with dementia to have experience of seeing a clinical pharmacist. However, due to challenges in recruiting participants with experience, we received ethical approval to change this to *any* primary care staff involved in their dementia care, for us to explore what they would *like* from a clinical pharmacist.

Carers were included if they were currently or previously had a role in caring for someone with a diagnosis of dementia, and who had experience of seeing a clinical pharmacist in their caring role.

Participant recruitment aimed to reflect the full trajectory of dementia. People with dementia who had capacity predominantly represented those in the earlier, milder stages of the condition, while carers providing perspectives relating to people in more advanced stages. This approach enabled us to capture a broad range of views across the dementia trajectory, acknowledging that care needs, including medical, nursing, and pharmaceutical, may vary significantly at different stages.

## Data collection

Two interview schedules guided the interviews (one for people with dementia, one for carers) (S1 Appendix). They were developed by the research team and all co-authors provided feedback, in collaboration with our Patient and Public Involvement and Engagement (PPIE) panel (MS and JWa). Our PPIE members provided feedback on all study documents, including piloting the interview schedules. Both interview schedules were modified iteratively as data generation and analysis of the transcripts progressed.

Interviews were conducted between December 2022 and April 2024 by telephone, video conferencing platforms (Microsoft Teams or Zoom), or in person, according to participant preferences. Informed written consent was obtained prior to the interviews, either in writing or audio recorded, and reconfirmed at the start and end of the interview. Interviews either took place in the remotely or the participant's home. Interviews were conducted by a female research assistant and female post-doctoral research fellow trained in qualitative research methods (AB and AW). Interviews were audio recorded and transcribed verbatim by a professional transcribing service. All interview transcripts were checked against the recordings by a researcher (AB) to ensure accuracy.

Demographic information, including age, ethnicity, level of education and career were collected via a questionnaire before the interview to contextualise the data and ensure we were recruiting to our sampling strategy.

At the end of the interviews, each participant completed a debrief form to understand their experience of the interview and assess any level of distress, as well as give them a chance to provide feedback. They were signposted to any services that were requested and were given the opportunity to receive the publication from the study. All participants were provided a £20 shopping voucher.

## Data analysis

The transcripts were uploaded and analysed using reflexive thematic analysis in NVivo 14 [35]. The research team represented a mix of professional backgrounds, all of whom had experience in qualitative research, social sciences and thematic analysis, and included GPs, pharmacists, psychology, and social care researchers. AB and ND initially coded four transcripts independently (two people with dementia and two carers) and met to discuss and agree codes. The resulting

coding framework was discussed with the whole team, each co-author providing feedback. AB coded all remaining interviews with regular discussions and input from ND and AW to refine and develop new codes. Themes were generated from the coding and discussed among the whole team, iteratively refined, and defined.

### Ethical approval

Ethical approval was obtained from the Health Research Authority (HRA) and Health and Care Research Wales (23/LO/0054) and University College London Research Ethics Committee (3344/006). Participants with dementia were deemed to have capacity to consent as per the Mental Capacity Act 2005 [36].

## Results

The study included 13 people with dementia and 15 carers. The people with dementia ranged from 56 to over 76 years of age, with the majority (n = 7) being between 66 and 75 years old, while carers ranged from 46 to over 76 years old. Both groups had a higher number of female participants (n = 8 people with dementia and n = 11 carers). Most participants with dementia were married (n = 8) and lived with family (n = 9), while most carers were married (n = 6) and lived in owner-occupied housing (n = 10). Alzheimer's was the main dementia type (n = 10), and time since diagnosis varied. Interviews with people with dementia lasted between 20–50 minutes and 30–60 minutes with carers. Thirteen participants with dementia and 10 carers completed their interview via video conferencing software, and the additional 3 carers completed their interview over the telephone. The participant groups were interviewed separately and were not dyads, except for two participants with dementia whose carers were present during the interview and were then retrospectively consented. These two additional carers did not complete the demographic questionnaires nor were they included in the carer transcript analysis. Six participants with dementia and 10 carers had experience of seeing a clinical pharmacist.

For full sample characteristics see Table 1.

Three overarching themes, each with subthemes, were developed from the interviews: 1) Recognising the value of clinical pharmacists; 2) Building and developing rapport, with personalised care and a holistic approach; and 3) The needs of patients without clinical pharmacy support – left in uncertainty about their care. An overview of the themes and subthemes can be seen in Table 2.

### Recognising the value of clinical pharmacists

Participants described both the benefits and limitations of clinical pharmacy services. While pharmacists were often valued for their expertise in managing medications and identifying interactions, dementia-specific support in their role was less consistent. Two sub-themes illustrate these perspectives 1) Dedicated medication reviews and long-term condition support by a clinical pharmacist; and 2) Limitations and awareness of the clinical pharmacy service.

**Dedicated medication reviews and long-term condition support by a clinical pharmacist.** Clinical pharmacists played a key role in medication reviews. These reviews aimed to assess medication adherence and optimisation and give people with dementia and their family carers an understanding of how, why and when to take medications and how they worked. Pharmacists were particularly valued for identifying potential medication interactions, which participants felt were sometimes overlooked by GPs. By clarifying complex medication information, clinical pharmacists acted as crucial intermediaries between patients/carers and other healthcare professionals:

*"So, my mum's on a different amount of medications, partly for Alzheimer's and then partly with other health things like blood pressure and stuff like that. And I guess a lot of people, unless you are in healthcare, don't necessarily think about the implications of different medications and their interactions and stuff. And it's something that I've asked my stepdad, and when I've been involved, helping with my mum, sometimes it's things like the GP hadn't noticed that the medications couldn't be together, particularly if she's been unwell." [CR15]*

**Table 1. Participant demographics.**

| | | People with dementia (n=13) | Carers (n=15) |
|---|---|---|---|
| Age | No response | 0 | 2 |
| | 18–35 | 0 | 2 |
| | 46–55 | 0 | 2 |
| | 56–65 | 1 | 5 |
| | 66–75 | 7 | 3 |
| | 76+ | 5 | 1 |
| Gender | Female | 5 | 11 |
| | Male | 8 | 4 |
| Marital Status | Single | 1 | 4 |
| | Married | 8 | 6 |
| | Divorced | 2 | 0 |
| | Separated | 0 | 2 |
| | Widowed | 1 | 1 |
| Residence | Private home (living alone) | 4 | 0 |
| | Private home (living with family or others) | 9 | 0 |
| | Council- rented housing | 0 | 1 |
| | Housing-association rented housing | 0 | 2 |
| | Private rented housing | 0 | 1 |
| | Owner-occupied housing | 0 | 10 |
| Sexual orientation | Heterosexual | 11 | 13 |
| | Bisexual | 0 | 1 |
| | Prefer not to say | 1 | 1 |
| Ethnicity | English | 10 | 9 |
| | Other White | 0 | 1 |
| | African | 1 | 0 |
| | Caribbean | 1 | 0 |
| | Indian | 0 | 1 |
| | Pakistani | 0 | 1 |
| | Other Asian | 0 | 1 |
| | Black British | 0 | 1 |
| | Arab | 0 | 1 |
| Level of education | No qualifications | 4 | 1 |
| | O levels | 2 | 3 |
| | A levels or other post O levels | 3 | 1 |
| | Degree | 1 | 4 |
| | Postgraduate | 2 | 5 |
| | Other | 1 | 1 |
| Religious background | Church of England | 4 | 4 |
| | Catholic | 1 | 1 |
| | No religious background | 7 | 3 |
| | Methodist | 1 | 1 |
| | Agnostic | 0 | 1 |
| | Humanist | 0 | 1 |
| | Spiritual | 0 | 1 |
| | Hindu | 0 | 1 |
| | Muslim | 0 | 1 |

*(Continued)*

**Table 1.** (Continued)

| | | People with dementia (n = 13) | Carers (n = 15) |
|---|---|---|---|
| Vocation | Technical and Creative Roles | 4 | 6 |
| | Healthcare and Support Roles | 2 | 4 |
| | Administrative and Professional Roles | 5 | 5 |
| | Other | 2 | 0 |
| Type of dementia | Alzheimer's disease | 10 | 0 |
| | Frontotemporal dementia | 1 | 0 |
| | Vascular dementia | 2 | 0 |
| Year since diagnosis | Less than 1 year | 4 | 0 |
| | 1–2 years | 2 | 0 |
| | 2–3 years | 1 | 0 |
| | 3–4 years | 5 | 0 |
| | More than 5 years | 1 | 0 |

While clinical pharmacists generally provided support for medication management which included medication for dementia, the extent of support offered varied. Management of symptoms associated with other long-term conditions often took priority, with dementia support secondary:

*"[…] as I'm talking to you, I'm realising that, that it was probably more with the co-morbidities that they helped me with to be honest with you. The actual dementia, they did to a degree, because obviously she was on the drug that she was, which name I've forgotten, she was on that which was – but at the end of the day, I know things are changing now and hopefully down the line it will be very different. There isn't a lot of medication for dementia. It tends to be the co-morbidities that you treat, isn't it? You know, it's the fact mum was falling, mum used to get infection, things like that, they're the ones she [the clinical pharmacist] helped with." [CR10]*

Some reported unmet expectations, describing instances where a medication review was expected, but did not occur:

*"She went into care and her diagnosis was vascular dementia, or whatever, it would have been nice at that time, I sup-pose, to have sat down with a clinical pharmacist who would then say, "Right, this is the medication that your mum is on and this is what we feel that she still needs" and the reason behind it." [CR08]*

**Limitations and awareness of the clinical pharmacy service.** Many people with dementia and carers were unclear about the role of clinical pharmacists, and how to access the service, with some unclear they had even seen a clinical pharmacist:

*"I saw the clinical pharmacist, although I didn't know that that person was a clinical pharmacist, but from your discus-sion with me over these weeks I realise that's the service that we had there." [CR03]*

Some perceived that pharmacists had a lack of clinical 'power' and authority. For example, not being able to always make changes to medications without the GP approval was sometimes seen as a limitation:

*"But never at once did the clinical pharmacist review my mum's medication, he didn't have enough power or authority, he said, "That would have to be consulted with the GP." So, I don't feel like that was useful. The clinical pharmacist should have reviewed the medication or should have been able to change it on the system." [CR07]*

**Table 2. Summary of themes.**

| Theme | Sub-themes | Summary |
|---|---|---|
| Recognising the value of clinical pharmacists | Dedicated medication reviews and long-term condition support by a clinical pharmacist<br>Limitations and awareness of the clinical pharmacy service | This theme highlights the recognition of the clinical pharmacist's value in managing medications, where participants emphasised the importance of timely, knowledgeable support and the continuity of care that pharmacists provided, particularly during transitions such as hospital discharge. This support helped reduce gaps in care and ensured better medication management for patients. |
| Building and developing rapport, with personalised care and a holistic approach | Personalised care<br>Communication and language | This theme highlights the positive impact of building rapport through personalised and holistic care provided by clinical pharmacists. Participants benefited from longer, more meaningful interactions with clinical pharmacists, compared to GPs, that allowed for tailored medication reviews and a deeper understanding of their care. Pharmacists were seen as central to offering a person-centred approach, addressing communication challenges, and ensuring patients understood their treatment, especially for those with cognitive impairments or language barriers. |
| The needs of patients without clinical pharmacy support - left in uncertainty about their care | | This theme underscores the unmet needs of patients without access to clinical pharmacy support, where participants described feeling uncertain about their care, particularly after diagnosis or hospital discharge, due to the lack of follow-up and clear guidance on medication management. |

However, this experience and perspective was not universal, again highlighting variation of the role across practices. Some participants recognised the unique role of clinical pharmacists in medication management and appreciated the distinct expertise they offered compared to GPs:

*"I think interactions between the different drugs is very important. And she's [clinical pharmacist] very good at that because whenever she arranges a telephone appointment with me, I always ask her, "I'm taking this, that, so on, so forth. My mum takes this, that. Do they interact with each other?" She knows; that's where I think the knowledge of the clinical pharmacist is even better than the doctors actually, sometimes." [CR11]*

Overall, there was little awareness and information about clinical pharmacy services, which compounded access to support:

*"Well I mean the thing is, which I think I've mentioned already, is that the definition "clinical pharmacist" is not, does not seem to be one that people are familiar with, even in the chemist. I went to our local chemist, I might have said this before, and I asked the assistant there, you know, "Is the chemist here a clinical pharmacist?" And she didn't know. And I think there is an NHS site which does define a clinical pharmacist but very few people seem aware of that." [PL05]*

Participants suggested that clearly defining and promoting the role of clinical pharmacists would improve public understanding and access to the service. This would help highlight the unique skills clinical pharmacists bring to the multidisciplinary primary care team:

*"I think it would be good if people knew specifically what the clinical pharmacist covers. To have someone there that would be easier to discuss a specialised subject that he is interested in, rather than trying to get through to the doctor, who you can't get through to anyway, most of the time. To know that someone there – it's like a physiotherapist. You know you go to a physiotherapist because of the problems you've got that need a physiotherapist. So, if people understood more what the clinical pharmacist could offer, to me it was just doing the prescriptions." [CR01]*

In summary, while clinical pharmacists were seen as medication experts, their potential to support dementia-specific needs was often underutilised due to lack of awareness, role clarity, and system-level inconsistency.

### Building and developing rapport, with personalised care and a holistic approach

For participants who had seen a clinical pharmacist, their accounts stood in sharp contrast to those who had not. This theme captures the positive impact of having a dedicated professional who provided tailored medication reviews, addressed concerns, and supported informed decisions. The first sub-theme explores personalised care, where longer and personalised appointments fostered trust and continuity. The second sub-theme highlights the critical role of communication particularly in dementia care, highlighting how clinical pharmacists' ability to adapt to communication needs was central to building rapport and delivering holistic, person-centred care. These experiences emphasised how pharmacists could fill a crucial gap in dementia care by providing proactive, person-centred support that others in the system often could not.

**Personalised care.** Clinical pharmacists were reported to complement and build upon services delivered by GPs. Longer appointments facilitated a holistic approach to medication reviews, resulting in a less transactional process. Clinical pharmacists often conducted home visits and maintained regular contact with patients and carers, enabling rapport building and continuity of care:

*"I think there was that element of personalised care really, which was nice. I think sometimes when you interact with the NHS, you're – for appointments with consultants and things, you're very much a number. You're very much kind of – you're in and out and their eyes are on the clock, and you've got to get out. "Right, are you happy with it? Right, off you go." It's kind of like that really. Whereas she [clinical pharmacist] did take quite a bit more time with us really. And it just felt like she cared a bit more, and that she was willing to get involved in it a little bit more than we'd had before." [CR14]*

Participants appreciated the extra time they were able to spend with the clinical pharmacist, compared to the short appointment times with GPs. The ability of clinical pharmacists to build rapport and adopt a holistic approach may allow them to function as effective intermediaries, gathering comprehensive information from patients and carers to inform the broader care team's understanding and decision-making:

*"We got into conversation and then she (clinical pharmacist) started asking me a lot of questions about myself and this and that and the other. So, she was actually – which is something which doesn't normally happen. You go in there and say, "This is my problem." And they (GPs) say, "OK, I'll prescribe these pills," and off you go. And it was far more than that. It was really, really good. I came back and told my wife and said, "I can't believe it. I've spent 35 minutes at the doctor (clinical pharmacist)." It's a bit unusual, isn't it?". [PL03]*

**Communication and language.** Participants expressed a preference for in-person reviews as it gave the clinical pharmacist the ability to visually and holistically assess the patient as well as include them in consultations. This was particularly important for those who may have communication difficulties and cognitive impairment, when communicating over the phone presented challenges:

*"So, yes, I think that that would be, I think they should get to know the person they're helping, rather than going through – even with the telephone, my husband couldn't talk to them on the phone. He just wouldn't, he'd say, "No, you do it." They're not seeing the person they're being asked to help, and I think that's very important, to assess, visually, what the person's like." [CR01]*

Some participants highlighted the importance of person-centred care and knowing the individual, with the ability to provide tailored communication. Some participants noted that people with dementia may not always be able to be included in consultations, especially if conducted over the phone, and the clinical pharmacist's approach should be adapted accordingly:

*"They (the clinical pharmacists and health care professionals) know my husband's got dementia, and it's my mobile but every time somebody rings, they'll say, "Can I speak to John?" and I say, "Well you can but I don't know what benefit you're going to get out of speaking to John." Basically, it's protocol that they're ringing about John, but it just amazes me that they still think my husband's got capacity to speak to them." [CR05]*

Effective communication was important to ensure the patient fully understood their treatment and instructions. One participant reported the clinical pharmacists was able to provide clarity and improve comprehension by simplifying complex medical terms:

*"Well, regarding the first visit – sorry, the first telephone consultation, I felt – I thought they – he (the clinical pharmacist) did a really good job. And he explained in non-medical jargon what the medication was for, which I found... A lot of – because they're healthcare professionals, they do speak in healthcare lingo. And a lot of – well, I don't understand a lot of it. So, it was helpful that they spoke in layman's terms." [PL02]*

Language differences impacted effective communication. Communication between those from different ethnic backgrounds and for non-English speakers was reported as a challenge when seeing a clinical pharmacist:

*"Whenever I've taken my mother into spaces like another surgery or another practice where she has an appointment [with a clinical pharmacist] for some other physical condition. And I've had to act as an interpreter during an appointment and I've questioned why is it that you did not get an interpreter? Did you not know that she isn't able to speak English? It should have been a question asked, "Does your mother require an interpreter?" [CR07]*

Barriers were also highlighted for those who live alone, especially with cognitive impairment, who may not have a family member, carer, or representative to advocate for them. Those individuals are at a significant disadvantage in accessing services and communicating with health care professionals such as clinical pharmacists:

*"No, the thing is sometimes as we are talking now, give me five minutes and I've forgotten what we were talking about or something like that, and that's a bit of a worry. The doctor might say, "Do this, do that," and I won't remember. So that's living; every time I'm going, I have to take my daughter with me. Then she will remember, knowing, "You're supposed to do this," so that's how I work it." [PL01]*

In summary, pharmacist support was described not just as a clinical interaction, but as a relationship that offered reassurance, continuity, and a sense of being seen.

## The needs of patients without clinical pharmacy support – left in uncertainty about their care

Despite mixed opinions and varying awareness of clinical pharmacists, the potential value of the role was underscored by the unmet needs of those who had not accessed clinical pharmacy support. This theme captures the experiences of participants who felt left without sufficient help to manage medications, particularly after diagnosis or hospital discharge. Many described a lack of follow-up, unclear advice, and difficulty navigating primary care, leaving them feeling uncertain and anxious about their treatment.

Without access to clinical pharmacists for medication reviews and deprescribing, patients often felt abandoned and lacked a dedicated point of contact for medication-related queries. Their accounts highlight the intermediary role that pharmacists could play in bridging communication gaps and providing specialised medication expertise -a role that is currently missing for many:

*"Nope we've not had any [support from services for dementia care]– once she was diagnosed, because she was an outpatient at a memory clinic. And then they (memory clinic) just said, "OK, we're going to discharge her. If you have any issues, go back to your GP, and then he will rerefer you back to us." And that's been it. So, they're just keeping an eye on her other medical conditions, but in terms of the dementia, there's been nothing." [Carer with PL09].*

Many patients continued to take medications initiated by memory clinics or hospital teams that were not routinely reviewed by GPs, leading to risk of polypharmacy. Those who had not seen a clinical pharmacist often said they lacked any regular discussions about their medicines:

*Interviewer: "Do you speak to the doctor or a chemist about these medications at all?"*

*Respondent: "Not really, no. They just tell you what they send you and then it's a – on repeat prescription and that. I do have a blood test once a year for my thyroid, because – obviously to check that and that. But that's it really, just leave you on them." [PL08]*

Medication changes made during hospital admissions also added complexity. Without pharmacist input post-discharge, risks such as interactions or duplications went unaddressed. In contrast, when pharmacists were involved, participants recalled clear and proactive communication:

*"I don't think initially [risks were explained], and I think that would have been when she was discharged from the hospital. But I know then when my stepdad went to get the repeat prescription, he'd then been contacted [by the clinical pharmacist] to say like, "Why is she now on – what changes this and this?" And he was then told [by the clinical pharmacist] about how she couldn't take, I think it was, one of her blood pressure medicines with one of the new antibiotics or antivirals that she'd been given (cont…) because I think she'd come out of hospital and was on a lot more than she'd gone in on." [CR15].*

Many participants expressed uncertainty about the necessity of certain medications or the potential benefits of alternative treatment options. Some reported feeling left in limbo about their medications with a lack of support around related queries:

*"You carry on taking the medication for months and months on end and you don't know where you are going and whether there is anything else that is coming on. No one suggests to me we have got something else [a medication], you are at the stage where you could benefit from something. There is nothing like that, you are just left in a vacuum, you take your pills every day and be quiet sort of thing." [PL13]*

Participants without clinical pharmacy input felt overlooked and unsupported in managing medications, particularly during care transitions. They described a lack of clear information, follow-up, and opportunities to ask questions, often feeling abandoned once specialist services stepped back. These experiences underscored the potential role of the clinical pharmacist in offering continuity, reassurance, and clearer lines of communication.

## Discussion

To our knowledge, this study is the first to explore the perspectives of people with dementia and family carers about the role of clinical pharmacists and support they receive in primary care. We have developed three themes from the interviews

that demonstrate both the benefits and challenges associated with clinical pharmacist care provision, offering insights to improve services and quality of care for people with dementia.

Participants who had engaged with clinical pharmacists described varied experiences. While many reported positive interactions, including valuable support with medication queries and optimisation; inconsistencies in the level of care provided were evident. Research has demonstrated clinical pharmacists improve patient safety through medication reviews and addressing polypharmacy [23] however the current research suggests the detail provided in medication reviews and signposting to external services varies significantly. Variation among healthcare professionals in dementia care has also been demonstrated in previous research. A study demonstrated factors associated with variability in care delivered by nurses and recommendations to reduce it, such as educating staff, standardising practice, including advanced care planning, and addressing systemic health inequities [37].

A common thread in our research was the lack of awareness of the clinical pharmacy service by carers and people with dementia, especially among participants who did not have access to these services. Some participants recognised the potential for clinical pharmacists to prescribe, review medications, and offer ongoing support to both patients and carers. However, others often lacked this understanding. This has also been found in previous research analysing the role of pharmacy independent prescribers (PIPs) who are based in care homes, which reported confusion among patients regarding the PIP service and were unsure if they could prescribe. This led to a preference for medication changes to remain with their GP [28,38]. This highlights that for clinical pharmacists to provide effective care to patients, as well as reduce GP workload, there is a need for enhanced clarity about their service they provide to the public.

People with dementia and family carers valued the ability of clinical pharmacists to provide clear communication as well as the ability to build trust and rapport. Establishing these strong therapeutic relationships with both people with dementia and their carers is paramount for healthcare professionals working in dementia care. A recent scoping review highlighted the importance of trust, clear communication, avoiding medical jargon, involving carers, and collaborating with the wider care team when clinicians are providing care for people with dementia [39]. Our research highlighted the potential role of clinical pharmacists in facilitating communication among people with dementia, their carers, and the wider primary care team. By acting as a central point of contact, clinical pharmacists could alleviate the burden on carers, who are often tasked with coordinating care and finding information from multiple healthcare providers. Carers frequently report having to proactively seek information, often encountering difficulties in obtaining timely and comprehensive updates [40]. Clinical pharmacists have the potential to act as intermediaries and enhance care coordination. Moreover, having additional time with the clinical pharmacist was seen as a benefit to patients and their carers, which may have facilitated the ability to develop trust and rapport. Extended time dedicated to people with dementia during clinic visits has been highly valued in previous research [41].

Participants who had not engaged with a clinical pharmacist expressed a need for a dedicated healthcare professional to address medication-related questions. Previous literature has also highlighted how patients and family carers often have a limited understanding of their medications and the side effects [42] and particularly in the context of dementia, can face many challenges in medication adherence and management without external support [43]. Clinical pharmacists have the expertise to bridge gaps in care by acting as key intermediaries, providing ongoing support and addressing medication and dementia-related concerns. In the current study many participants also reported feeling isolated following a dementia diagnosis and a subsequent lack of ongoing support from primary care services. Research has demonstrated inequalities in access to support services, including post-diagnostic care and commencing dementia medication [44,45]. Again, clinical pharmacists may be able to fulfil this intermediary role by providing on-going medication support for people with dementia, as well as complete referrals and signpost to external services.

## Implications for research, policy and practice

Our findings highlight a need for support among individuals who have not accessed the clinical pharmacist services. Clinical pharmacists are well positioned to address the specific needs of this patient group, such as the ability to provide

essential medication management and enhance communication between all parties involved in their care. Conversely, while some patients have benefited from clinical pharmacist involvement, there are variations in care provision, such as the availability of the service, adding to a postcode lottery of care that many people with dementia report for a number of care services [46]. Prioritising and incorporating dementia-specific training, guidelines, and specialised support may improve these variations. Future research should also explore clear role definitions for clinical pharmacy services for dementia care, to reduce these variations in service delivery across England.

It is important to increase this awareness amongst the public to enhance engagement, cohesion and understanding that clinical pharmacists have wider contributions to patient care, including medication changes.

## Strengths and limitations

We were able to recruit a large and diverse sample in terms of age, sex and education, which included 28 participants. Using semi-structured interviews, we were able to explore in-depth family carer's and people with dementia's experiences and perspectives on clinical pharmacist and primary care support. However, the sample of participants were solely from England and so may not be representative of other healthcare systems worldwide as well as other parts of the UK that also have clinical pharmacists in primary care. Moreover, the sample was largely White English, representing limited ethnic diversity. Despite there being no report of difficulties, the online format and reliance on video conferencing software may have introduced barriers for the participants living with dementia. In-person interviews might offer a more supportive and accessible environment for this population.

## Conclusion

This study offers novel insights into how people with dementia and family carers perceive dementia support within primary care in England, particularly focusing on clinical pharmacists. Our findings reveal a need for improving access to clinical pharmacist services across the UK, as evidenced by the unmet support needs of those who have not accessed them. While some participants had positive experiences, inconsistencies in care provision were prevalent. To optimise the role of clinical pharmacists in dementia care, increasing public awareness, and further research into effective service delivery models are essential.

## Supporting information

**S1 Appendix.  DCPharm topic guide - Carers.**
(DOCX)

## Author contributions

**Conceptualization:** Alice Burnand, Abi Woodward, Nathan Davies.

**Data curation:** Alice Burnand, Abi Woodward.

**Formal analysis:** Alice Burnand, Abi Woodward, Nathan Davies.

**Funding acquisition:** Nathan Davies.

**Investigation:** Alice Burnand.

**Methodology:** Alice Burnand, Nathan Davies.

**Project administration:** Alice Burnand.

**Resources:** Alice Burnand.

**Software:** Alice Burnand.

**Supervision:** Nathan Davies.

**Writing – original draft:** Alice Burnand.

**Writing – review & editing:** Alice Burnand, Abi Woodward, Kumud Kantilal, Cini Bhanu, Yogini Jani, Jill Manthorpe, Mine Orlu, Greta Rait, Madiha Sajid, Kritika Samsi, Victoria Vickerstaff, Jane Ward, Jane Wilcock, Nathan Davies.

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
