## [Decision Letter · Decision Letter 0]

21 Apr 2025

PONE-D-24-58779Understanding the perspectives of people with dementia and family carers about clinical pharmacists in primary care: a qualitative studyPLOS ONE

Dear Dr. Burnand,

Thank you for submitting your manuscript to PLOS ONE. After careful consideration, we feel that it has merit but does not fully meet PLOS ONE’s publication criteria as it currently stands. Therefore, we invite you to submit a revised version of the manuscript that addresses the points raised during the review process. The paper has been reviewed by two expert in the field, who see the principal value of the paper, but require clarification mostly concerning methodological issues.

We look forward to receiving your revised manuscript.

Kind regards,

Sascha Köpke

Academic Editor

PLOS ONE

Journal Requirements:

“Alice Burnand is supported by the National Institute for Health and Care Research (NIHR) School for Primary Care Research (project reference 575).

Abigail Woodward is supported by the National Institute for Health and Care Research (NIHR) School for Primary Care Research Capacity Award (grant reference C093).

Kumud Kantilal is supported by the National Institute for Health and Care Research ARC North Thames. The views expressed are those of the author(s) and not necessarily those of the National Institute for Health and Care Research or the Department of Health and Social Care.

Cini Bhanu is supported by the National Institute for Health and Care Research (NIHR) School for Primary Care Research.

Yogini Jani is supported by the UCLH NHS Foundation Trust and secondment to UCL.

Jill Manthorpe is supported by Kings College London and the National Institute for Health and Care Research (NIHR).

Mine Orlu is supported by the National Institute for Health and Care Research (NIHR) School for Primary Care Research.

Greta Rait is supported by grant and infrastructure from the National Institute for Health and Care Research (NIHR).

Madiha Sajid is supported by the National Institute for Health and Care Research (NIHR) School for Primary Care Research (project reference 575).

Kritika Samsi is supported by the National Institute for Health and Care Research (NIHR).

Victoria Vickerstaff is supported by the National Institute for Health and Care Research (NIHR) School for Primary Care Research (project reference 575).

Jane Ward is supported by the National Institute for Health and Care Research (NIHR) School for Primary Care Research (project reference 575).

Jane Wilcock is supported by the National Institute for Health and Care Research (NIHR) School for Primary Care Research (project reference 575).

Nathan Davies is supported by the National Institute for Health and Care Research (NIHR) School for Primary Care Research (project reference 575).”

” This study/project is funded by the National Institute for Health and Care Research (NIHR) School for Primary Care Research (project reference 575). The views expressed are those of the authors and not necessarily those of the NIHR or the Department of Health and Social Care. The funders had and will not have a role in study design, data collection and analysis, decision to publish, or preparation of the manuscript.”

 “Alice Burnand is supported by the National Institute for Health and Care Research (NIHR) School for Primary Care Research (project reference 575).

Abigail Woodward is supported by the National Institute for Health and Care Research (NIHR) School for Primary Care Research Capacity Award (grant reference C093).

Kumud Kantilal is supported by the National Institute for Health and Care Research ARC North Thames. The views expressed are those of the author(s) and not necessarily those of the National Institute for Health and Care Research or the Department of Health and Social Care.

Cini Bhanu is supported by the National Institute for Health and Care Research (NIHR) School for Primary Care Research.

Yogini Jani is supported by the UCLH NHS Foundation Trust and secondment to UCL.

Jill Manthorpe is supported by Kings College London and the National Institute for Health and Care Research (NIHR).

Mine Orlu is supported by the National Institute for Health and Care Research (NIHR) School for Primary Care Research.

Greta Rait is supported by grant and infrastructure from the National Institute for Health and Care Research (NIHR).

Madiha Sajid is supported by the National Institute for Health and Care Research (NIHR) School for Primary Care Research (project reference 575).

Kritika Samsi is supported by the National Institute for Health and Care Research (NIHR).

Victoria Vickerstaff is supported by the National Institute for Health and Care Research (NIHR) School for Primary Care Research (project reference 575).

Jane Ward is supported by the National Institute for Health and Care Research (NIHR) School for Primary Care Research (project reference 575).

Jane Wilcock is supported by the National Institute for Health and Care Research (NIHR) School for Primary Care Research (project reference 575).

Nathan Davies is supported by the National Institute for Health and Care Research (NIHR) School for Primary Care Research (project reference 575)”

6. Please include your tables as part of your main manuscript and remove the individual files. Please note that supplementary tables (should remain/ be uploaded) as separate "supporting information" files

Reviewers' comments:

Reviewer's Responses to Questions

**Comments to the Author**

1. Is the manuscript technically sound, and do the data support the conclusions?

Reviewer #1: Yes

Reviewer #2: Partly

2. Has the statistical analysis been performed appropriately and rigorously? 

Reviewer #1: N/A

Reviewer #2: N/A

3. Have the authors made all data underlying the findings in their manuscript fully available?

Reviewer #1: Yes

Reviewer #2: Yes

4. Is the manuscript presented in an intelligible fashion and written in standard English?

Reviewer #1: Yes

Reviewer #2: Yes

5. Review Comments to the Author

Reviewer #1: Thank you very much for the opportunity to read this interesting paper about pharmacists‘ services for patients with dementia.

In most (if not all) countries health services are some sort of restricted by financial constraints. This becomes particularly obvious for patients who are vulnerable and do have special needs. Moreover, patients living with dementia require caregiver, who are frequently overcharged.

However, to understand the role of clinical pharmacists and what they can contribute it is also mandatory, to understand the healthcare system. In this case it would be helpful fort he reader to know, how the NHS is financed and as I understand, the GP (who does not have enough time, why?) is included in the NHS services. And the pharmacists seem to have more time – but how is their time paid for? CG 14 on page 17 seems to understand, that the pharmacist is not part of the NHS?

In addition, what is the difference between a pharmacist and a clinical pharmacist in the UK? For example, in Germany the one who has a private pharmacy (where patients get their medications and advice) can specialise to become a clinical pharmacist, who is allowed after this special training to make medication reviews which are financed by the statutory health fund. This service is paid for with approx. 90,- € per medication review.

All together, the themes evolved and discussed show the lack of general service, e.g. patient-centered care, support between sectors (hospital care – day clinics – living at home) and less a problem of medications.

It would be helpful to go more into the details of a) what could pharmacists offer and b) what else is expected from the NHS.

There are some unclear aspects: are clinical pharmacists allowed to prescribe/deprescribe like some nurse practitioners?

The presenstation of the interviews is nicely embedded in the results section, however, the authors may think about a more systematic presentation by themes and sub-themes, requirements and potential solutions. As it is done in the paper by Sawan et al on a very similar topic, namely qualitative interviews with CGs about pharmacy services for patients with dementia after hospital discharge. This paper is very similar and should be considered for discussion.

Sawan MJ, Jeon YH, Bond C, Hilmer SN, Chen TF, Wennekers D, Gnjidic D. Caregivers' experiences of medication management advice for people living with dementia at discharge. J Eval Clin Pract. 2021 Dec;27(6):1252-1261. doi: 10.1111/jep.13551.

Reviewer #2: Peer Review

Summary:

This qualitative study explores the perspectives of individuals with dementia and their carers regarding the role and involvement of clinical pharmacists in primary care teams. Data were collected via semi-structured interviews and analyzed using reflexive thematic analysis. The paper presents three themes related to pharmacists' roles, their influence on care, and the absence of their involvement.

General Assessment:

The paper addresses a timely and relevant topic and is generally well written. However, several aspects require clarification and refinement to enhance its scientific rigor and transparency. It is unclear whether COREQ or SRQR guidelines were followed. The reference formatting, punctuation, and citation order require correction. The contribution of the large author group should be briefly explained.

Introduction

• The role of clinical pharmacists within the care team should be more clearly described. Do they work alongside general practitioners (GPs) or independently? How are they integrated into the team? How do patients access them?

• Additional contextualization is needed: What is already known about patients’ and carers’ perspectives on clinical pharmacists across various conditions? Why is it particularly important to explore the perspectives of people with dementia and their carers? Do the responsibilities of clinical pharmacists differ when working with this population? Given the limited pharmacological options for dementia treatment, what role do pharmacists play

Research Design

• Recruitment: The recruitment strategy and inclusion criteria need further elaboration, particularly regarding geographical diversity and participant selection. Also, clarify which social media platforms were used for advertising. Since a purposive sampling strategy was applied, how were participants actively selected?

• Inclusion Criteria: Was previous contact with a clinical pharmacist an inclusion criterion for both patients and carers? The range from mild to severe dementia entails significant differences in medical, nursing, and pharmaceutical care, which warrants further justification for the broad inclusion.

• More detail is needed on the development of the interview schedule and the involvement of PPIE. Including the interview guides as a table or supplement is recommended.

• The duration of the interviews and the composition of participant groups should be reported in the results section.

Results

• The themes in the results section lack clear structure and explanation. A table or figure summarizing themes and subthemes would aid comprehension. Themes should be introduced before detailing subthemes.

• There is some inconsistency in the focus—at times it is on the clinical pharmacist, and at other times on the broader care team. This should be addressed in the introduction to the themes or clarified throughout.

• The title of the first theme—“understanding”—is ambiguous. Does it refer to the researchers’ understanding of participants’ perspectives or participants’ perceived usefulness of clinical pharmacists? Would “perceived usefulness” alone suffice? Clarifying the focus and expanding the theme to include the broader context of clinical pharmacists’ roles within care teams would strengthen the discussion. If this broader perspective is not addressed, it should be noted, with examples to illustrate the focus.

• In the second subtheme, the third quote (“I think interactions between…”) appears to align more closely with the first subtheme. Please clarify its placement or consider moving it.

• The second theme begins with references to the clinical pharmacist but quickly shifts to more general language issues. This transition should be clarified or refocused.

• The third theme raises broader concerns. Some quotes appear to come from participants without pharmacist contact, which contradicts the stated inclusion criteria. Clarification is needed regarding timing and context of these experiences.

Discussion

• Some information presented in the discussion for the first time (PIPs) should be moved to the results section and later be discussed here.

• The potential role of pharmacists as intermediaries in dementia care teams is compelling but needs more emphasis in the results.

• From my view, interviews suggest that care teams without clinical pharmacists do not fully meet participants' needs, and that interprofessional collaboration remains a challenge even with their involvement. Why is an effective care team particularly important for individuals with dementia? The potential for pharmacists to serve as intermediaries warrants further exploration.

• While the issue of access is mentioned in the discussion, it is not addressed in the introduction and appears only briefly in the results. Greater emphasis on access throughout the paper would be beneficial.

• The limited diversity of the sample should be discussed as a potential limitation.

6. PLOS authors have the option to publish the peer review history of their article (what does this mean? ). If published, this will include your full peer review and any attached files.

**Do you want your identity to be public for this peer review?** For information about this choice, including consent withdrawal, please see our Privacy Policy .

Reviewer #1: No

Reviewer #2: No

---

## [Author Response · Author response to Decision Letter 1]

6 May 2025

Editors’ comments Response

1. Please ensure that your manuscript meets PLOS ONE's style requirements, including those for file naming. The PLOS ONE style templates can be found at …

Thank you, the requirements have now been met.

Please include your amended Funding Statement within your cover letter. We will change the online submission form on your behalf. Thank you, this has been added into the manuscript and cover letter.

Please confirm at this time whether or not your submission contains all raw data required to replicate the results of your study.

Yes, this is correct.

These have been corrected and added into the cover letter.

” This study/project is funded by the National Institute for Health and Care Research (NIHR) School for Primary Care Research (project reference 575). The views expressed are those of the authors and not necessarily those of the NIHR or the Department of Health and Social Care. The funders had and will not have a role in study design, data collection and analysis, decision to publish, or preparation of the manuscript.”

“Alice Burnand is supported by the National Institute for Health and Care Research (NIHR) School for Primary Care Research (project reference 575).

Abigail Woodward is supported by the National Institute for Health and Care Research (NIHR) School for Primary Care Research Capacity Award (grant reference C093).

Kumud Kantilal is supported by the National Institute for Health and Care Research ARC North Thames. The views expressed are those of the author(s) and not necessarily those of the National Institute for Health and Care Research or the Department of Health and Social Care.

Cini Bhanu is supported by the National Institute for Health and Care Research (NIHR) School for Primary Care Research.

Yogini Jani is supported by the UCLH NHS Foundation Trust and secondment to UCL.

Jill Manthorpe is supported by Kings College London and the National Institute for Health and Care Research (NIHR).

Mine Orlu is supported by the National Institute for Health and Care Research (NIHR) School for Primary Care Research.

Greta Rait is supported by grant and infrastructure from the National Institute for Health and Care Research (NIHR).

Madiha Sajid is supported by the National Institute for Health and Care Research (NIHR) School for Primary Care Research (project reference 575).

Kritika Samsi is supported by the National Institute for Health and Care Research (NIHR).

Victoria Vickerstaff is supported by the National Institute for Health and Care Research (NIHR) School for Primary Care Research (project reference 575).

Jane Ward is supported by the National Institute for Health and Care Research (NIHR) School for Primary Care Research (project reference 575).

Jane Wilcock is supported by the National Institute for Health and Care Research (NIHR) School for Primary Care Research (project reference 575).

Nathan Davies is supported by the National Institute for Health and Care Research (NIHR) School for Primary Care Research (project reference 575)”

Please include the funding statement only as the one stated in the cover letter, including these additional statements: Abigail Woodward is supported by the National Institute for Health and Care Research (NIHR) School for Primary Care Research Capacity Award (grant reference C093).

Kumud Kantilal is supported by the National Institute for Health and Care Research ARC North Thames. The views expressed are those of the author(s) and not necessarily those of the National Institute for Health and Care Research or the Department of Health and Social Care.

6. Please include your tables as part of your main manuscript and remove the individual files. Please note that supplementary tables (should remain/ be uploaded) as separate "supporting information" files. Thank you this has now been added in.

Reviewer #1 comments Author’s response

Thank you very much for the opportunity to read this interesting paper about pharmacists‘ services for patients with dementia.

In most (if not all) countries health services are some sort of restricted by financial constraints. This becomes particularly obvious for patients who are vulnerable and do have special needs. Moreover, patients living with dementia require caregiver, who are frequently overcharged.

However, to understand the role of clinical pharmacists and what they can contribute it is also mandatory, to understand the healthcare system. In this case it would be helpful for the reader to know, how the NHS is financed and as I understand, the GP (who does not have enough time, why?) is included in the NHS services. And the pharmacists seem to have more time – but how is their time paid for? CG 14 on page 17 seems to understand, that the pharmacist is not part of the NHS?

Thank you for highlighting this. We have added detail about how the NHS is financed in the UK (line 136-138). We have also included detail about GP workload (line 132-134) and strain explaining why they have less time. We are focussing on the role of clinical pharmacists as part of the NHS.

In addition, what is the difference between a pharmacist and a clinical pharmacist in the UK? For example, in Germany the one who has a private pharmacy (where patients get their medications and advice) can specialise to become a clinical pharmacist, who is allowed after this special training to make medication reviews which are financed by the statutory health fund. This service is paid for with approx. 90,- € per medication review.

An explanation of the difference between community and clinical pharmacists is on line 140-145.

All together, the themes evolved and discussed show the lack of general service, e.g. patient-centered care, support between sectors (hospital care – day clinics – living at home) and less a problem of medications.

It would be helpful to go more into the details of a) what could pharmacists offer and b) what else is expected from the NHS.

Thank you this is an important point. The aim of this paper was to understand the perspective of carers and people with dementia on the services that are currently provided. However, we have weaved this in throughout the results through the significant revisions made as much as possible from the data we have.

There are some unclear aspects: are clinical pharmacists allowed to prescribe/deprescribe like some nurse practitioners?

Thank you, this has been added in introduction (line 144-145).

The presentation of the interviews is nicely embedded in the results section; however, the authors may think about a more systematic presentation by themes and sub-themes, requirements and potential solutions. As it is done in the paper by Sawan et al on a very similar topic, namely qualitative interviews with CGs about pharmacy services for patients with dementia after hospital discharge. This paper is very similar and should be considered for discussion.

Sawan MJ, Jeon YH, Bond C, Hilmer SN, Chen TF, Wennekers D, Gnjidic D. Caregivers' experiences of medication management advice for people living with dementia at discharge. J Eval Clin Pract. 2021 Dec;27(6):1252-1261. doi: 10.1111/jep.13551.

Thank you for highlighting this useful reference. We have made significant revisions to the results section as per this comment and those of Reviewer #2’s to enhance clarity and comprehension of the themes.

This paper has been included in the introduction, thank you for providing this useful reference (line 159-161).

Summary:

This qualitative study explores the perspectives of individuals with dementia and their carers regarding the role and involvement of clinical pharmacists in primary care teams. Data were collected via semi-structured interviews and analyzed using reflexive thematic analysis. The paper presents three themes related to pharmacists' roles, their influence on care, and the absence of their involvement.

General Assessment:

The paper addresses a timely and relevant topic and is generally well written. However, several aspects require clarification and refinement to enhance its scientific rigor and transparency. It is unclear whether COREQ or SRQR guidelines were followed. The reference formatting, punctuation, and citation order require correction. The contribution of the large author group should be briefly explained.

Thank you for your comments. We have added in about our use of the COREQ checklist (line 194).

Contributions have been added and references amended.

Reviewer #2 comments Author’s response

Introduction

• The role of clinical pharmacists within the care team should be more clearly described. Do they work alongside general practitioners (GPs) or independently? How are they integrated into the team? How do patients access them?

Thank you this has been added in introduction (line 151-152).

• Additional contextualization is needed: What is already known about patients’ and carers’ perspectives on clinical pharmacists across various conditions? Why is it particularly important to explore the perspectives of people with dementia and their carers? Do the responsibilities of clinical pharmacists differ when working with this population? Given the limited pharmacological options for dementia treatment, what role do pharmacists play

Thank you for raising these comments. They have been addressed in the introduction section regarding other literature on perspectives of clinical pharmacists (line 157-159), their role in managing polypharmacy/supporting people with dementia and why it is important to conduct this study (line 176-184).

Research Design

• Recruitment: The recruitment strategy and inclusion criteria need further elaboration, particularly regarding geographical diversity and participant selection. Also, clarify which social media platforms were used for advertising. Since a purposive sampling strategy was applied, how were participants actively selected?

This has been added into the methods section (line 198-207).

• Inclusion Criteria: Was previous contact with a clinical pharmacist an inclusion criterion for both patients and carers? The range from mild to severe dementia entails significant differences in medical, nursing, and pharmaceutical care, which warrants further justification for the broad inclusion.

Thank you, these points have been addressed in the methods section page 9 and 10. Lines 227-230 and lines 236-241.

• More detail is needed on the development of the interview schedule and the involvement of PPIE. Including the interview guides as a table or supplement is recommended.

Thank you, more detail has been added (line 247-248) and the interview schedules added in the appendix.

• The duration of the interviews and the composition of participant groups should be reported in the results section.

This has been moved from the methods and added to the results section (line 294-295).

Results

• The themes in the results section lack clear structure and explanation. A table or figure summarizing themes and subthemes would aid comprehension. Themes should be introduced before detailing subthemes.

We have made significant revisions to the results throughout for improved clarity, explanation and structure. A description and explanation of the themes has been added for clarity before introducing subthemes. We have reorganised the themes and ensured the thread across the themes is clearer. Finally, we have added table 2 - a summary of the themes and sub-themes.

• There is some inconsistency in the focus—at times it is on the clinical pharmacist, and at other times on the broader care team. This should be addressed in the introduction to the themes or clarified throughout.

Thank you, included in the introductions to each theme are the details regarding the focus.

• The title of the first theme—“understanding”—is ambiguous. Does it refer to the researchers’ understanding of participants’ perspectives or participants’ perceived usefulness of clinical pharmacists? Would “perceived usefulness” alone suffice? Clarifying the focus and expanding the theme to include the broader context of clinical pharmacists’ roles within care teams would strengthen the discussion. If this broader perspective is not addressed, it should be noted, with examples to illustrate the focus.

Thank you for highlighting this. We have renamed this theme ‘Recognising the value of clinical pharmacists’ and enhanced clarity within the introduction to the theme.

• In the second subtheme, the third quote (“I think interactions between…”) appears to align more closely with the first subtheme. Please clarify its placement or consider moving it.

Thank you for this comment. This is now theme 3 due to our reorganisation to improve flow.

We provided this quote to show the variety of perceptions of the role of the clinical pharmacist, highlighting confusion about their responsibilities. This lack of clarity sometimes led to unmet expectations, including feeling that clinical pharmacists lacked the authority to make necessary medication changes independently (previous quote on line 370). The subsequent quote you have highlighted on line 379 shows not all participants felt this, and many were aware that they had a key and independent role from doctors. Therefore, we believe this quote has relevance to stay within this subtheme.

• The second theme begins with references to the clinical pharmacist but quickly shifts to more general language issues. This transition should be clarified or refocused.

We have provided more clarity in the introduction to this theme. This is now theme 3.

• The third theme raises broader concerns. Some quotes appear to come from participants without pharmacist contact, which contradicts the stated inclusion criteria. Clarification is needed regarding timing and context of these experiences.

Thank you, the inclusion criteria has been expanded to enhance clarity about the inclusion criteria for people with dementia who provided the quotes for this theme (line 222).

Discussion

• Some information presented in the discussion for the first time (PIPs) should be moved to the results section and later be discussed here.

Apologies for the confusion on this. The term ‘the study’ was about the literature from previous research about PIPs. We have changed the wording to improve clarity. See line 604-607.

• The potential role of pharmacists as intermediaries in dementia care teams is compelling but needs more emphasis in the results.

Thank you, this had been added further throughout the results section about the role of the intermediary.

From my view, interviews suggest that care teams without clinical pharmacists do not fully mee

---

## [Decision Letter · Decision Letter 1]

14 Jul 2025

PONE-D-24-58779R1Understanding the perspectives of people with dementia and family carers about clinical pharmacists in primary care: a qualitative studyPLOS ONE

Dear Dr. Burnand,

Thank you for submitting your manuscript to PLOS ONE. After careful consideration, we feel that it has merit but does not fully meet PLOS ONE’s publication criteria as it currently stands. Therefore, we invite you to submit a revised version of the manuscript that addresses the points raised during the review process.

We look forward to receiving your revised manuscript.

Kind regards,

Sascha Köpke

Academic Editor

PLOS ONE

Journal Requirements:

Reviewers' comments:

Reviewer's Responses to Questions

**Comments to the Author**

1. If the authors have adequately addressed your comments raised in a previous round of review and you feel that this manuscript is now acceptable for publication, you may indicate that here to bypass the “Comments to the Author” section, enter your conflict of interest statement in the “Confidential to Editor” section, and submit your "Accept" recommendation.

Reviewer #1: All comments have been addressed

Reviewer #2: All comments have been addressed

2. Is the manuscript technically sound, and do the data support the conclusions?

Reviewer #1: (No Response)

Reviewer #2: Partly

3. Has the statistical analysis been performed appropriately and rigorously? 

Reviewer #1: (No Response)

Reviewer #2: N/A

4. Have the authors made all data underlying the findings in their manuscript fully available?

Reviewer #1: (No Response)

Reviewer #2: Yes

5. Is the manuscript presented in an intelligible fashion and written in standard English?

Reviewer #1: (No Response)

Reviewer #2: Yes

6. Review Comments to the Author

Reviewer #1: (No Response)

Reviewer #2: Dear Authors,

Thank you very much for your revision.

The manuscript has improved substantially, especially the well-written introduction now provides a solid overview of the topic.

I have just a few remaining comments:

Methods

1. Page 9, line 232: Is "clinician" the correct term here? The revision is unclear—please consider clarifying or rephrasing.

2. How was the interview guide developed? Was it based on literature, and did you use an inductive or deductive approach? Was a pilot interview conducted? I could not locate the interview schedule in the version I received—possibly a formatting issue?

3. Please indicate how many participants were interviewed via phone, video, or in person. Remote interviews with people living with dementia can be challenging. If in-person interviews were not feasible, please explain and discuss this as a possible limitation, ideally supported by relevant literature.

Results

1. Some participants (how many?) had no prior contact with clinical pharmacists. This could be also mentioned later as a limitation, although it may have helped address recruitment challenges.

2. I think about reordering the themes: starting with the current Themes 1 and 3 would better reflect your research questions. Theme 2, based on a broader perspective, might fit better at the end.

3. Is it standard in your analysis approach to include brief descriptions of sub-themes in the main theme titles? From my experience, this is not always necessary if sub-themes are described in the text.

Discussion

1. I suggest focusing strongly on Themes 1 and 3 in the discussion. Theme 2 appears more speculative and might be best addressed at the end.

Thank you very much for your work.

7. PLOS authors have the option to publish the peer review history of their article (what does this mean? ). If published, this will include your full peer review and any attached files.

**Do you want your identity to be public for this peer review?** For information about this choice, including consent withdrawal, please see our Privacy Policy .

Reviewer #1: **Yes: ** Petra A. Thürmann

Reviewer #2: No

---

## [Author Response · Author response to Decision Letter 2]

22 Jul 2025

Reviewer’s comments and Author response

Methods

1. Page 9, line 232: Is "clinician" the correct term here? The revision is unclear—please consider clarifying or rephrasing.

Thank you for highlighting this, this has been changed to enhance clarity.

2. How was the interview guide developed? Was it based on literature, and did you use an inductive or deductive approach? Was a pilot interview conducted? I could not locate the interview schedule in the version I received—possibly a formatting issue?

Lines 245-250 explain how the interview guide was developed. The Interview Schedule was also added as Appendix 1.

3. Please indicate how many participants were interviewed via phone, video, or in person. Remote interviews with people living with dementia can be challenging. If in-person interviews were not feasible, please explain and discuss this as a possible limitation, ideally supported by relevant literature.

This has been added into the results section lines 298-301. It has also been added as a limitation lines 659-662.

Results

1. Some participants (how many?) had no prior contact with clinical pharmacists. This could be also mentioned later as a limitation, although it may have helped address recruitment challenges.

This has been added into the results section 304-305. However, we do not see this to be a limitation as the majority of participants had experience of a clinical pharmacist, and we were able to explore from the additional participants what support they need and how a clinical pharmacist could help.

2. I think about reordering the themes: starting with the current Themes 1 and 3 would better reflect your research questions. Theme 2, based on a broader perspective, might fit better at the end.

Thank you for your suggestion, the themes have been reordered.

3. Is it standard in your analysis approach to include brief descriptions of sub-themes in the main theme titles? From my experience, this is not always necessary if sub-themes are described in the text.

This was added in our previous revision in response to reviewer feedback, which we think provides greater clarity and guidance for the reader.

Discussion

1. I suggest focusing strongly on Themes 1 and 3 in the discussion. Theme 2 appears more speculative and might be best addressed at the end.

Thank you, the discussion has been adapted to reflect this. We have also added in additional research to further support themes (1) and (2).

---

## [Editor Report · Decision Letter 2]

25 Jul 2025

Understanding the perspectives of people with dementia and family carers about clinical pharmacists in primary care: a qualitative study

PONE-D-24-58779R2

Dear Dr. Burnand,

We’re pleased to inform you that your manuscript has been judged scientifically suitable for publication and will be formally accepted for publication once it meets all outstanding technical requirements.

Kind regards,

Sascha Köpke

Academic Editor

PLOS ONE